# Effects of acupuncture on age-related macular degeneration: A systematic review and meta-analysis of randomized controlled trials

Wu Sun[1,2☯], Yuwei Zhao[1☯], Liang Liao[3], Xueyao Wang[4], Qiping Wei[3], Guojun Chao[4]*, Jian Zhou[iD][3]*

1 Beijing University of Chinese Medicine, Beijing, China, 2 Xiyuan Hospital of China Academy of Chinese Medical Sciences, Beijing, China, 3 Dongfang Hospital, Beijing University of Chinese Medicine, Beijing, China, 4 Eye Hospital, Chinese Academy of Chinese Medical Sciences, Beijing, China

☯ These authors contributed equally to this work.
* chaoguojun@sohu.com (GC); dfyk55555@126.com (JZ)

## Abstract

### Background

In recent years, an increasing number of patients with age-related macular degeneration (AMD) have received acupuncture treatment, but there has been no systematic review to evaluate the effect of acupuncture on patients with AMD.

### Purpose

This meta-analysis aims to review the clinical efficacy of acupuncture in the treatment of AMD.

### Methods

Randomized controlled trials up to September 4, 2022 were searched in the following databases: PubMed, Ovid Medline, Embase, Cochrane Library, The Chinese National Knowledge Infrastructure Database, VIP, Wanfang, and SINOMED. Two reviewers independently performed literature screening and data extraction. RevMan 5.4 was used for the meta-analysis.

### Results

Nine of the 226 articles were finally included. A total of 508 AMD patients (631 eyes) were enrolled, including 360 dry eyes and 271 wet eyes. The results showed that acupuncture alone or as an adjunct therapy improved both the clinical efficacy and best-corrected visual acuity (BCVA) of AMD patients and reduced their central macular thickness. The certainty of the evidence ranged from "low" to "very low".

**Data Availability Statement:** All data can be found in the article.

**Funding:** This study was supported by The National Natural Science Foundation of China

(NSFC), No.81874491. The funders had no role in study design, data collection and analysis, decision to publish, or preparation of the manuscript.

**Competing interests:** The authors have declared that no competing interests exist.

## Conclusion

There is no high-quality evidence that acupuncture is effective in treating patients with AMD; patients with dry AMD may benefit from acupuncture treatment. Considering the potential of acupuncture treatment for AMD, it is necessary to conduct a rigorously designed randomized controlled trials to verify its efficacy.

## 1. Introduction

Age-related macular degeneration (AMD) is a disease that affects the macular area of the retina and causes progressive damage to central vision [1]. The early stage manifests as drusen and retinal pigment epithelium abnormalities, and the later stage manifests as dry AMD characterized by non-vascular proliferation or wet AMD characterized by choroidal neovascularization (CNV). Later, AMD can cause loss of central visual acuity of the patient, accompanied by severe visual impairment and even blindness [2]. Currently, AMD is the leading cause of irreversible blindness in developed countries [3, 4]. A meta-analysis involving a total of 129,664 participants worldwide showed that the overall global prevalence of early- and late-stage AMD in the adult population was 8.01% and 0.37%, respectively, with a total prevalence of 8.69% [5]. Considering that aging is the greatest risk factor for AMD, the number of people suffering from AMD will continue to increase [6]. By 2040, the number of AMD patients is expected to increase to 288 million [5].

The current treatment measures for AMD include anti-vascular endothelial growth factor (VEGF) injection therapy, antioxidant vitamins and minerals, photodynamic therapy, thermal laser photocoagulation surgery, etc [7, 8]. To date, there is no effective treatment for dry AMD patients other than appropriate antioxidant supplementation. Anti-VEGF therapy, the current first-line treatment, may reduce the odds of legal blindness caused by neovascular AMD [9]. However, a long-term follow-up study of patients initially receiving regular anti-VEGF agents showed that among patients who were followed up for more than 7 years, two-thirds lost most of their gains in visual acuity (VA) [10]. Currently, an increasing number of other treatment measures are causing concern. Relevant studies have shown that acupuncture may become an effective method for the treatment of AMD, which can improve the eye symptoms and visual acuity of patients [11, 12]. There are the following hypotheses about the mechanism of acupuncture treatment for AMD. Previous research revealed that acupuncture has obvious specificity for the macular area [13]. Acupuncture could improve microcirculation in the macular area by expanding the surrounding blood vessels after acting on the peripheral area of the eyes [14–16]. Clinical studies have further confirmed that acupuncture can reduce central macular thickness (CMT) and promote the absorption of fundus exudate [16–18]. In addition, relevant study have also shown that acupuncture may reduce serum VEGF levels in patients with wet AMD [19, 20]. Although many studies have revealed an intervention effect of acupuncture on AMD, there has been no review to scientifically evaluate the efficacy of acupuncture therapy. This review aims to evaluate the efficacy and safety of acupuncture in the treatment of AMD.

## 2. Methods

### 2.1. Study registration

This meta-analysis was registered with the International Prospective Register of Systematic Reviews (PROSPERO; registration number: CRD42020168611) and strictly adhered to the Preferred Reporting Items for Systematic Reviews and Meta-analyses (PRISMA) [21].

## 2.2. Inclusion criteria

**2.2.1. Types of studies.** Randomized controlled trials (RCTs).

**2.2.2. Types of participants.** Patients diagnosed with AMD by the investigators of the original study. Dioagnostic criteria inlucding at least one of the following:

1) Presence of at least intermediate-size drusen ($\geq$63 μm in diameter); 2) Retinal pigment epithelium (RPE) abnormalities; 3) Presence of any of the following features: geographic atrophy of the RPE, CNV (wet), polyidal choroidal vasculopathy, reticular pseudodrusen, or retinal angiomatous proliferation [8].

**2.2.3. Types of interventions.** All types of acupuncture were included, including electroacupuncture, warm acupuncture, and scalp acupuncture. Interventions in the control group included sham acupuncture, no treatment, placebo, antioxidants or other treatments. To evaluate the efficacy of acupuncture as an adjunctive therapy, studies that compared acupuncture combined with other therapies with the other therapies were also included.

**2.2.4. Outcome measures.** The primary outcomes were clinical efficacy rates and BCVA. The clinical efficacy rates were defined as the number of patients who showed improvement in VA and the absorption of fundus bleeding and exudate, and the improvements were assessed by clinicians based on the "Criteria of Diagnosis and Therapeutic Effect of Internal Diseases and Syndromes in Traditional Chinese Medicine (TCM)" [22] (S1 File in S1 Appendix). The secondary outcomes included changes in the CMT of the patients and adverse events.

## 2.3. Exclusion criteria

Studies involving any of the following were not included: 1) research that involved a comparison of different acupuncture techniques or a comparison of different acupuncture points; 2) treatments involving laser needle or hydro-acupuncture therapy; 3) duplicate published studies or case reports.

## 2.4. Search strategy

Relevant literature was searched in the following databases: PubMed, Ovid Medline, Embase, Cochrane Library, The Chinese National Knowledge Infrastructure Database, VIP, Wanfang, and SINOMED. The search time was from inception to September 4, 2022. In addition, relevant web pages were also manually searched (www.clinicaltrials.gov; www.clinicaltrialsregister.eu; trialsearch.who.int) for ongoing trials or unpublished clinical trial reports. The specific search strategy can be found in S1 Table in S1 Appendix.

## 2.5. Data extraction

Two reviewers conducted a literature search independently. After screening out the duplicate documents in EndNote software, a preliminary review was carried out by reading the titles and abstracts of the retrieved documents. The literature that satisfied the inclusion and exclusion criteria was read in full to determine its eligibility for further inclusion. Any differences between the two reviewers were resolved through communication and negotiation with an arbiter.

## 2.6. Quality assessment

The methodological quality of the included studies was evaluated according to the Cochrane risk-of-bias tool for randomized trials (RoB 2.0) as follows [23]: randomization process, deviations from the intended interventions, missing outcome data, measurement of the outcome and selection of the reported result. Each item was classified as "low risk of bias", "some problems", and "high risk of bias".

## 2.7. Data analysis

RevMan 5.4 was used for the meta-analysis. Continuous outcome variables were calculated by mean differences (MDs) or standard mean differences (SMDs) with 95% confidence intervals (CIs), and dichotomous outcome variables were calculated by risk ratios (RRs) with 95% CIs. When the heterogeneity of outcome variables was low ($P > 0.10$, $I^2 < 50\%$), the fixed-effect model was used; otherwise, the random-effect model was used. Publication bias assessment based on funnel plots was performed when the number of included studies was greater than 10. Subgroup analysis was performed by intervention type (with or without TCM), AMD type (dry or wet), or intervention course. Sensitivity analyses were performed to observe changes in synthetic results according to the following operations: 1) excluding low-quality studies; 2) excluding studies with small sample size; 3) excluding studies with the largest sample size; 4) excluding studies containing Chinese herbal medicine; 5) or switching between fixed and random effects models.

## 2.8. Quality of evidence

The quality of the pooled evidence for all the outcomes was judged by two independent reviewers according to the Grading of Recommendations Assessment, Development, and Evaluation (GRADE) system [24]. The strength of evidence was graded as "high", "moderate", "low" or "very low" based on five assessment items: risk of bias, inconsistency, indirectness, imprecision, and other considerations.

# 3. Results

## 3.1. Literature search

A total of 226 articles were included, of which 102 studies were removed due to duplication. After reading the titles and abstracts, 15 articles remained (110 articles were removed, including 68 irrelevant articles, 26 reviews, 15 case reports and case series, and 1 mechanistic study). Of these, 4 included inappropriate comparisons, 1 was a comparison of different acupuncture manipulation techniques, and 1 was a comparison of different acupoints. Finally, nine studies were included [15–18, 25–29] (Fig 1). Excluded studies were listed in S1 Table in S1 Appendix.

## 3.2. Characteristics of the studies

All 9 RCTs were conducted in China. A total of 508 AMD patients (631 eyes) were enrolled, including 360 eyes with dry AMD and 271 eyes with wet AMD. The study period was from 2011 to 2020, and the patients were over 50 years old. Of the 9 studies, 3 studies consisted of patients with wet AMD [15, 16, 18], 4 studies consisted of patients with dry AMD [25–28], and the remaining 2 studies included patients with dry and wet AMD [17, 29]. Three studies compared acupuncture with antioxidants (vitamin E/C) [18, 25, 29]. Five studies compared acupuncture combined with medication therapy with the same medication therapy, in which all the medications were Chinese herbs [15–17, 26, 27]. One study compared an acupuncture group with a control group, and the control group was only observed regularly without any treatment [28]. All the studies involved true acupuncture with filiform needles. The course of treatment ranged from 20 days to 3 months. In terms of outcome indicators, 7 studies reported clinical effectiveness [15–18, 26, 27, 29], 5 studies reported BCVA [16–18, 25, 27], and 3 studies mentioned CMT [16–18]. Side effects were reported in only 2 studies [17, 28]. While the BCVA included in all studies were tested using international standard visual acuity charts, the recording method used was the five-point recording method or decimal recording method. Tables 1 & 2 list the specific information from the studies included.

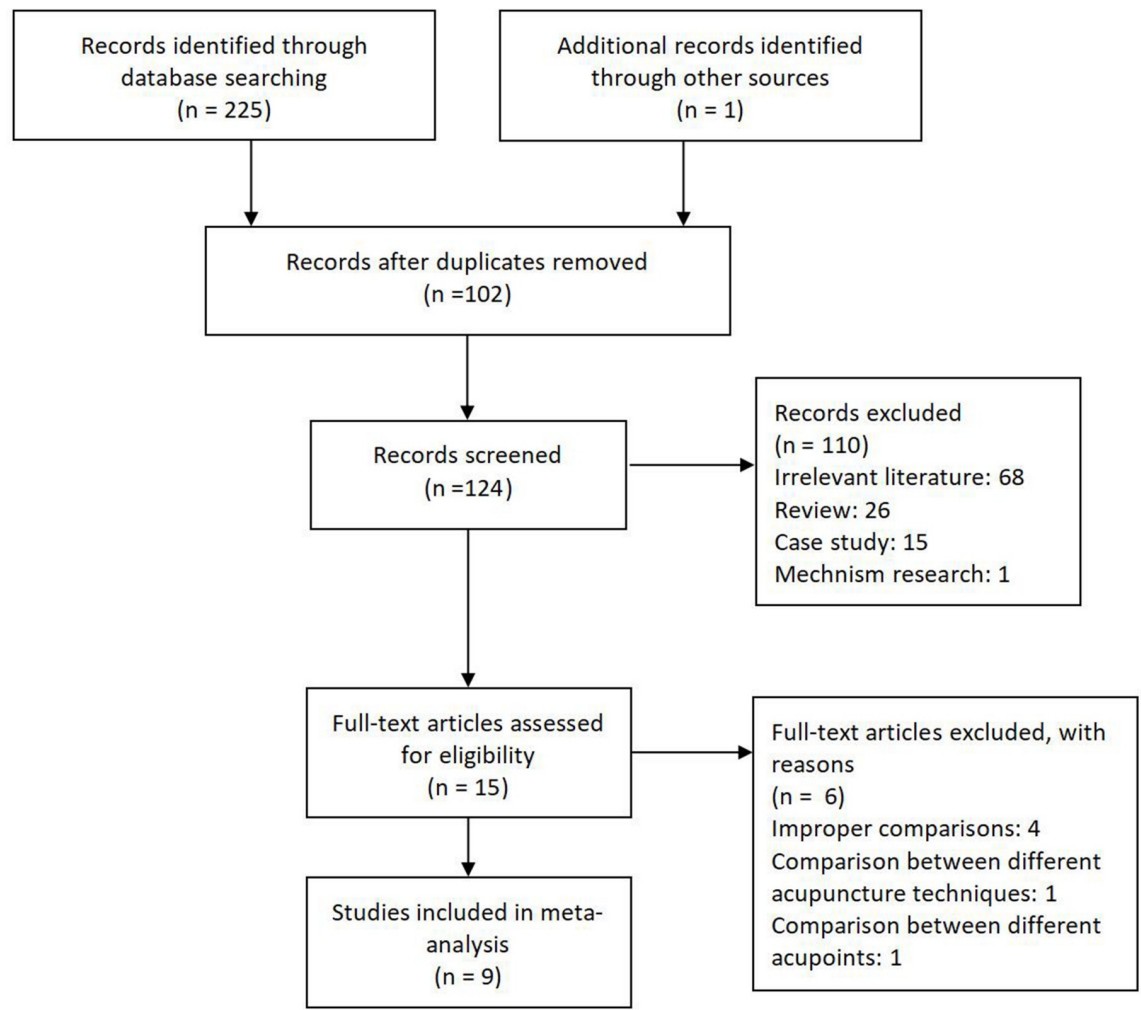

**Fig 1. Study flow diagram.**

**Table 1. Summary of included studies.**

| Study ID | Sample size (eyes) | | Age (years) (M/F) | | AMD type | Intervention | | Outcomes |
|---|---|---|---|---|---|---|---|---|
| | E | C | E | C | | E | C | |
| Yao 2015 [15] | 23 cases (36 eyes) | 22 cases (34 eyes) | 69.2±9.8 (27/18) | | Wet AMD | A+TCM | TCM | ① |
| Yang 2019 [16] | 34 cases (47 eyes) | 31 cases (40 eyes) | 63.62±8.56 (16/18) | 63.46±8. 65 (15/16) | Wet AMD | A+TCM | TCM | ①~③ |
| Wang 2019 [17] | 37 cases (37 eyes) | 37 cases (37 eyes) | 57.52±4.16 (24/13) | 57.55±4.11 (25/12) | Dry + wet AMD | A+TCM | TCM | ①~④ |
| liu 2016 [18] | 20 cases (20 eyes) | 19 cases (20 eyes) | 66.72±8.71 (10/10) | 65.24± 9.01 (11/8) | Wet AMD | A | Vitamin C/E | ①~③ |
| Xia 2014 [25] | 22 cases (44 eyes) | 15 cases (30 eyes) | 60±4 (12/10) | 61±3 (8/7) | Dry AMD | A | Vitamin C/E | ② |
| Wang 2017 [26] | 36 cases (36 eyes) | 36 cases (36 eyes) | 58.7±3.4 (30/42) | | Dry AMD | A+TCM | TCM | ① |
| Qin 2020 [27] | 30 cases (30 eyes) | 30 cases (30 eyes) | > 50 (NR) | | Dry AMD | A+TCM | TCM | ①② |
| Xia 2013 [28] | 22 cases (44 eyes) | 10 cases (20 eyes) | 60.45±3.85(12/10) | 60.56±3.23(4/6) | Dry AMD | A | No treatment | ④ |
| Jiao 2011 [29] | 56 cases (60 eyes) | 28 cases (30 eyes) | 50–80 (NR) | | Dry + wet AMD | A | Vitamin C/E | ① |

A, acupuncture; C, control group; E, experimental group; NR, not reported; TCM, traditional chinese medicine; ① clinical efficacy rates; ② best-corrected visual acuity (BCVA); ③ central macular thickness (CMT); ④ adverse events.

**Table 2. Descriptions of acupuncture interventions.**

| Study ID | Acupuncture stimulation method | Needle type | Acupuncture point | Retention time | Treatment regimen | Practitioner background |
|---|---|---|---|---|---|---|
| Yao 2015 [15] | MA | NR | BL1,EX-HN7,GB1, LR3,ST1,shangming, jianming | NR | 1 session every 2 days for 21 days | NR |
| Yang 2019 [16] | MA | NR | BL2,BL23,EX-HN5,EX-HN7,L14,L20,SP6, ST36 | 30min | 1 session every day for 3 months | NR |
| Wang 2019 [17] | MA | NR | BL1,EX-HN5,GB1, GB37, K16, L14, LI11,ST36,ST40 | 30min | 2 sessions a week for 2 months | NR |
| liu 2016 [18] | MA | 0.25×25 mm | BL1,BL2, EX-HN1, GB1 | 30min | 1 session every day for 30 days | NR |
| Xia 2014 [25] | MA | 0.25×(40–50) mm | BL1,BL2, EX-HN5, EX-HN14,GB20,ST1 | / | 2 session a week for 2 months | NR |
| Wang 2017 [26] | MA | 0.25×25 mm | BL1,BL2,BL24,BL26,CV9,DU20,EX-HN4,EX-HN5, EX-HN7,GB14,GB37,K13, LR3, SP6, SP10, ST1,ST36, ST40, shangming | 30min | 1 session every day for 30 days | NR |
| Qin 2020 [27] | MA | NR | BL10,EX-HN5,GB20,SJ23,ST1,xinming | 20min | 1 session every day for 20 days | NR |
| Xia 2013 [28] | MA | 0.25×(40/50) mm | BL1,EX-HN7, GB1, LR3, ST1 | 30min | 2 sessions a week for 2 months | NR |
| Jiao 2011 [29] | MA | 0.25×(25–50) mm | BL1,BL2,BL18,BL20,BL23,EX-HN5,GB1,GB14,GB20, GB37, ST2, ST40 | 30min | 1 session every day for 50 days | NR |

MA, manual acupuncture; NR, not reported.

## 3.3. Risk of bias assessment

All studies mentioned randomization, of which 4 studies used a random number table [15, 25, 28, 29] and 5 did not mention random methods. None of the studies mentioned allocation concealment. Considering that without sham acupuncture, the patients could not be blinded to the treatment, we think that blinded the acupuncture practitioner is equivalent to blinding. Three studies had a high risk of selective reporting [17, 26, 29]. All studies reported complete results with no significant missing data. Five studies were supported by government funding [15, 16, 18, 25, 28], and the remaining studies did not mention funding information. See Fig 2 for details.

## 3.4. Outcome measurements

**3.4.1. Clinical efficacy rates.** The clinical efficacy rates were evaluated in 7 studies [15–18, 26, 27, 29], and there was no heterogeneity in the outcome ($I^2 = 0\%$). The fixed-effect model showed that compared with the control, acupuncture significantly improved the clinical efficacy rates of the AMD patients (RR = 1.29, 95% CI: 1.17,1.42) (Fig 3A).

**3.4.2. BCVA.** BCVA was mentioned in 5 studies [16–18, 25, 27], with very high heterogeneity in the results ($I^2 = 89\%$). The random-effect model indicated that there was a significant difference between the acupuncture group and the control group in terms of BCVA (SMD = 0.95, 95% CI: 0.26,1.64) (Fig 3B).

**3.4.3 CMT.** CMT was mentioned in three studies [16–18], with high heterogeneity in the results ($I^2 = 67\%$). The random-effect model suggested that the acupuncture group had reduced CMT compared with that in the control group (MD = - 32.74, 95% CI: - 60.96,-4.55). (Fig 3C).

**3.4.4 Adverse events.** Adverse events were reported in only 2 studies [17, 28]. In one study, 10 of 22 patients in the acupuncture group had bleeding [28], while no side effects occurred in the control group. In another study, 3 patients in the control group and the acupuncture group each had nausea, vomiting, and flushing [17].

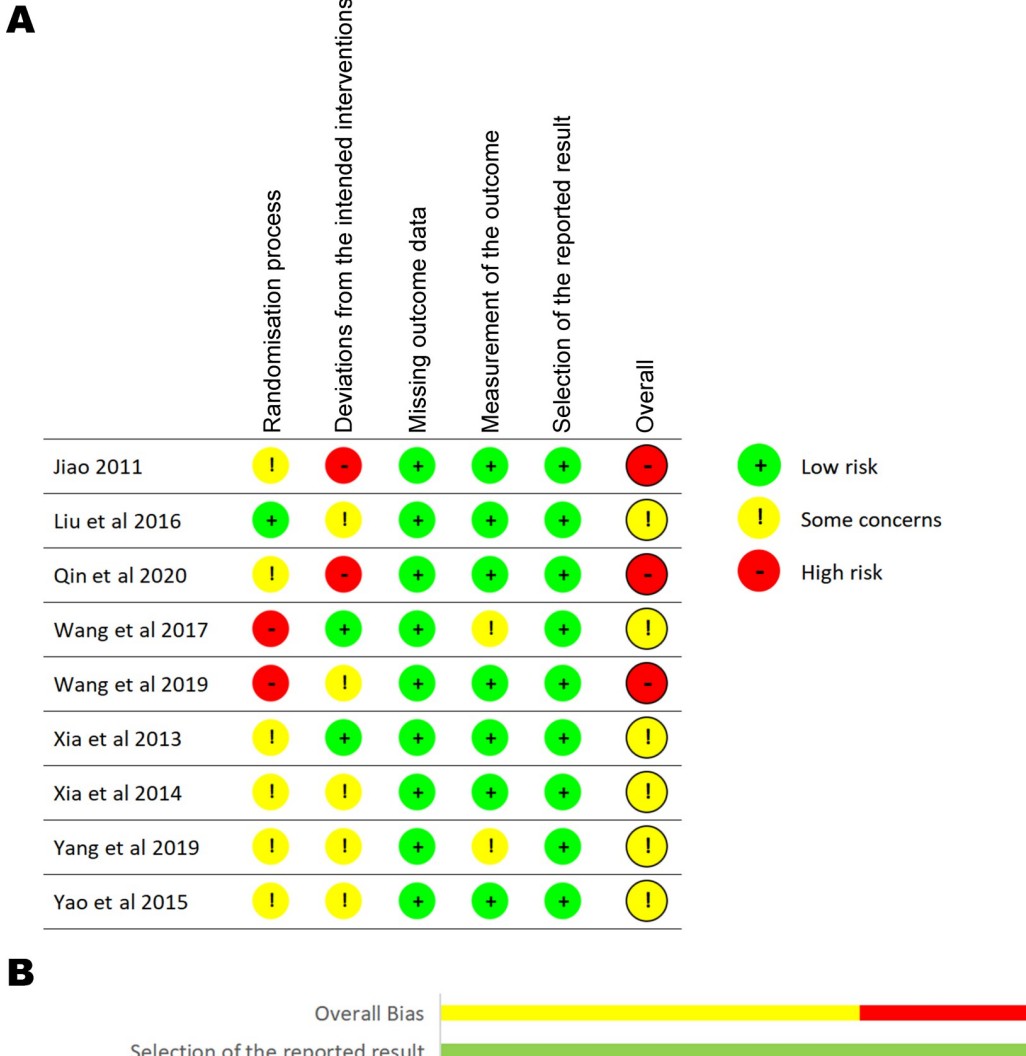

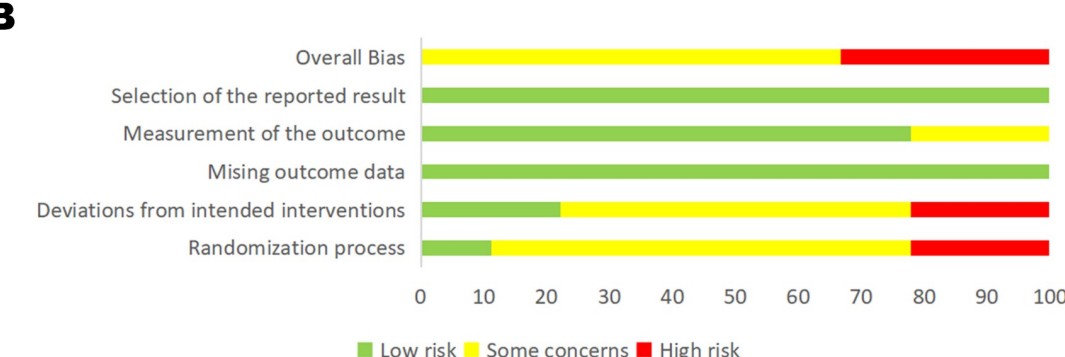

**Fig 2. Risk of bias assessment of included studies.** A, literature quality evaluation; B, summary of the quality evaluation of the literature.

**3.4.5 Sensitivity analysis and subgroup analysis.** Sensitivity analysis showed the stability of both Clinical efficacy rates and BCVA outcomes. When one study was removed [16], the difference in CMT between the two groups was no longer statistically significant.

Subgroup analysis showed that heterogeneity in BCVA and CMT outcomes disappeared when limiting the type of AMD to wet AMD, suggesting that heterogeneity was mainly related to the type of AMD. In addition, heterogeneity decreased to varying degrees when restricting

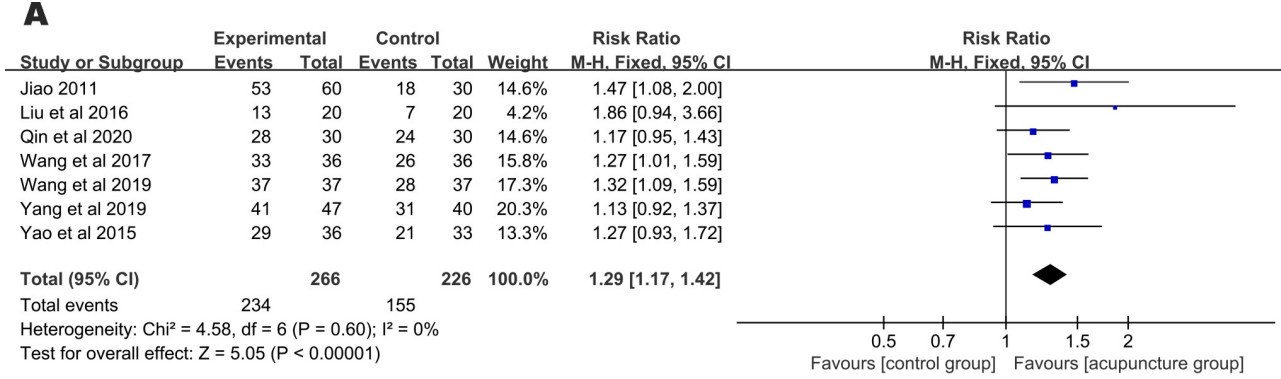

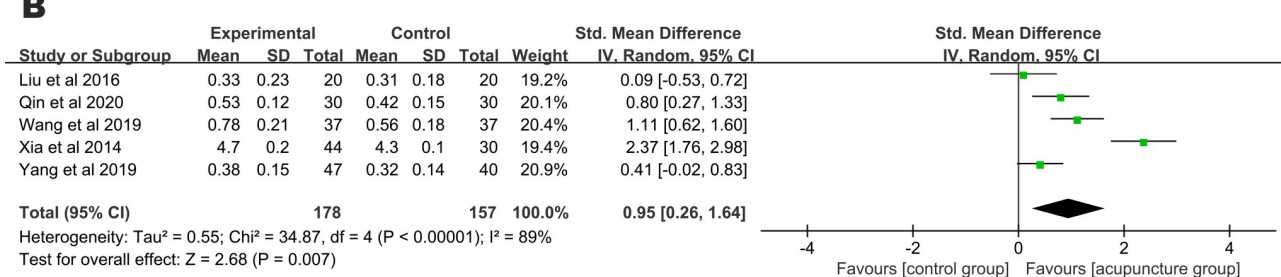

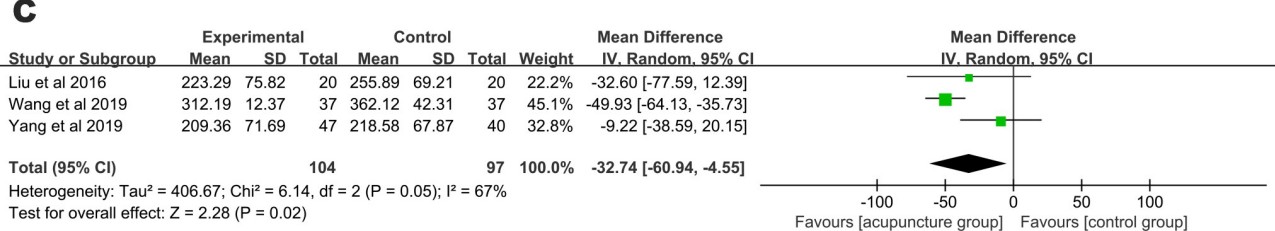

**Fig 3. Forest plot of meta-analysis results.** (A) Clinical response rates; (B) BCVA; (C) CMT.

the duration of the intervention (duration = 20–50 days) and the type of intervention (intervention includes TCM).

In the subgroup analysis of wet AMD or short intervention course (duration = 20–50 days), we noted changes in BCVA outcomes, with no statistically significant difference between the acupuncture group and the control group. See Tables 3 & 4 for details.

# 4. Discussion

## 4.1. Summary of evidence

The meta-analysis showed that compared with conventional treatment, acupuncture treatment increased the clinical efficacy and visual acuity of patients with AMD. In addition, acupuncture may have had a positive effect on the CMT of AMD patients. However, the certainty of the evidence was low due to concerns about the quality of the included studies.

According to TCM theory, inner eye tissue is closely related to meridians, which are the distribution network of essential substances of qi, blood, and body fluids throughout the body. The macula is yellowish on the unshaded fundus or eyeball, so it is considered to belong to the foot Taiyin Spleen Channel [30]. The spleen has the function of controlling the normal operation of blood in the meridians. Once the spleen-qi movement is not comfortable, it will impair

**Table 3. Subgroup analysis.**

| | Subgroup analysis | No. of trials (reference nos.) | No. of patients | $I^2$ | RR [95% CI] | P value |
|---|---|---|---|---|---|---|
| **Clinical efficacy rates** | Overall | 7 (15–18,24,25,27) | 492 | 0 | 1.29 [1.17, 1.42] | <0.0001 |
| | Type of intervention | | | | | |
| | Including TCM | 5 (15–17,24,25) | 362 | 0 | 1.22 [1.11, 1.35] | <0.0001 |
| | Without TCM | 2 (18,27) | 130 | 0 | 1.56 [1.17, 2.07] | 0.002 |
| | Intervention course | | | | | |
| | 20–50 days | 5 (15,18,24,25,27) | 331 | 0 | 1.33 [1.17, 1.52] | <0.0001 |
| | 51–90 days | 2 (16,17) | 161 | 21 | 1.21 [1.06, 1.39] | 0.006 |
| | AMD type | | | | | |
| | Dry AMD | 2 (24,25) | 132 | 0 | 1.22 [1.05, 1.42] | 0.01 |
| | Wet AMD | 3 (15,16,18) | 196 | 18 | 1.26 [1.06, 1.50] | 0.01 |
| | Dry and wet AMD | 2(17,27) | 164 | 0 | 1.39 [1.16, 1.65] | 0.0002 |
| | **Subgroup analysis** | **No. of trials (reference nos.)** | **No. of patients** | **$I^2$** | **SMD [95% CI]** | **P value** |
| **BCVA** | Overall | 5 (16–18,23,25) | 335 | 89% | 0.95 [0.26,1.64] | 0.007 |
| | Type of intervention | | | | | |
| | Including TCM | 3 (16,17,25) | 221 | 56% | 0.76 [0.34, 1.18] | 0.0004 |
| | Without TCM | 2 (18,23) | 114 | 96% | 1.23 [-1.00, 3.46] | 0.28 |
| | Intervention course | | | | | |
| | 20–50 days | 2 (18,25) | 100 | 65% | 0.47 [-0.22, 1.16] | 0.18 |
| | 50–90 days | 3 (16,17,23) | 235 | 93% | 1.28 [0.21, 2.34] | 0.02 |
| | AMD type | | | | | |
| | Dry AMD | 2 (23,25) | 134 | 93 | 1.58 [0.04, 3.11] | 0.04 |
| | Wet AMD | 2 (16,18) | 127 | 0 | 0.31 [-0.04, 0.66] | 0.09 |
| | Dry and wet AMD | 1 (17) | 74 | NA | 1.11 [0.62, 1.60] | <0.0001 |
| | **Subgroup analysis** | **No. of trials (reference nos.)** | **No. of patients** | **$I^2$** | **MD [95% CI]** | **P value** |
| **CMT** | Overall | 3(16–18) | 201 | 67% | -32.74 [-60.94, -4.55] | 0.02 |
| | Type of intervention | | | | | |
| | Including TCM | 2 (16,17) | 161 | 83% | -31.69 [-71.37, 7.99] | 0.12 |
| | Without TCM | 1 (18) | 40 | NA | -32.60 [-77.59, 12.39] | 0.16 |
| | Intervention course | | | | | |
| | 20–50 days | 1 (18) | 40 | NA | -32.60 [-77.59, 12.39] | 0.16 |
| | 51–90 days | 2 (16,17) | 161 | 83% | -31.69 [-71.37, 7.99] | 0.12 |
| | AMD type | | | | | |
| | Wet AMD | 2 (16,18) | 127 | 0 | -16.21 [-40.80, 8.39] | 0.2 |
| | Dry and wet AMD | 1 (17) | 74 | NA | -49.93 [-64.13, -35.73] | <0.0001 |

NA, not applicable.

the spleen's ability to control the blood flow, which will cause the macular blood to flow out of the blood vessels. Of the twelve channels, only the liver channel is directly connected to the eyes. Therefore, the liver channel plays an important role in connecting the eyes with the liver, which can communicate the flow of qi and blood between the two organs. In the TCM system, the liver is the main reservoir of blood and the eyes need to be nourished by blood. In addition, liver qi is also closely related to eye function. Only when the liver-qi is comfortable can the eyes perform optimally [30]. Therefore, Qi and blood are very important for the eyes, and regulating qi and blood has become an important principle in the treatment of AMD in TCM theory.

In TCM theory, acupoints are specific parts of the body surface that reflect the state of human organs and regulate their physiological functions. Through acupuncture to stimulate

**Table 4. Sensitivity analyses.**

| | Sensitivity analyses | No. of included trials (reference nos.) | RR [95% CI] | P |
|---|---|---|---|---|
| **Clinical efficacy rates** | Overall analysis | 7 (15–18,24,25,27) | 1.29 [1.17, 1.42] | <0.0001 |
| | Excluding studies with low quality | 6 (15,17,18,25,27) | 1.33 [1.19, 1.49] | <0.0001 |
| | Excluding small trial (participants < 50) | 6 (15–17,24,25,27) | 1.26 [1.15, 1.39] | <0.0001 |
| | Excluding the lagest trial | 6 (15,17,18,25,27) | 1.33 [1.19, 1.49] | <0.0001 |
| | Excluding studies containing TCM | 2 (18,27) | 1.56 [1.17, 2.07] | 0.002 |
| | Using random-effects model | 7 (15–18,24,25,27) | 1.25 [1.14, 1.37] | <0.0001 |
| | **Sensitivity analyses** | **No. of included trials (reference nos.)** | **SMD [95% CI]** | **P** |
| **BCVA** | Overall analysis | 5 (16–18,23,25) | 0.95 [0.26,1.64] | 0.007 |
| | Excluding studies with low quality | 4 (16–18,23) | 0.99 [0.09, 1.89] | 0.03 |
| | Excluding small trial (participants < 50) | 4 (16,17,23,25) | 1.15 [0.38, 1.92] | 0.003 |
| | Excluding the lagest trial | 3 (17,18,23,25) | 1.09 [0.24, 1.95] | 0.01 |
| | Excluding studies containing TCM | 3 (16,17,25) | 0.76 [0.34, 1.18] | 0.0004 |
| | Using fixed-effects model | 5 (16–18,23,25) | 0.88 [0.65, 1.12] | <0.0001 |
| | **Sensitivity analyses** | **No. of included trials (reference nos.)** | **MD [95% CI]** | **P** |
| **CMT** | Overall analysis | 3 (16–18) | -32.74 [-60.94, -4.55] | 0.02 |
| | Excluding studies with low quality | 2 (17,18) | -31.69 [-71.37, 7.99] | 0.12 |
| | Excluding small trial (participants < 50) | 2 (17,18) | -31.69 [-71.37, 7.99] | 0.12 |
| | Excluding the lagest trial | 2 (16,17) | -48.36 [-61.90, -34.81] | <0.0001 |
| | Excluding studies containing TCM | 1 (18) | -32.60 [-77.59, 12.39] | 0.16 |
| | Using fixed-effects model | 3 (16–18) | -41.49 [-53.79, -29.19] | <0.0001 |

the acupoint of the meridians, the human qi machinery is regulated, so as to ease the movement of qi and blood of the meridians, so that the eyes can get the nourishment of qi and blood, and finally treat the discomfort of the eyes. In addition, acting on acupoints around the eye is also believed to stimulate the movement of qi and blood in the eye and treat eye diseases. For example, the acupoint around the eyes, Jingming Point (BL1). As the first point of the bladder meridian, BL1 receives the ascending qi and blood from the bladder meridian and supplies it to the eyes. The eye receives the supply of qi and blood and therefore can see clearly. At present, acupuncture treatment of AMD mostly uses a combination of periocular and systemic acupuncture points [31].

Relevant histological studies have found that local acupoints may include high-density nerve endings, nerve and vascular components, and mast cells with sensory stimulation functions [32]. Acupuncture treatment of AMD mostly uses a combination of periocular and systemic acupuncture points. When the points are stimulated by acupuncture, in addition to the local release of biological factors to regulate local effects, acupuncture also transmits somatosensory information to the central nervous system by stimulating the nerves connected to the skin and muscles, thereby regulating the function of the autonomic nervous system [32, 33].

Oxidative stress is known to be one of the important pathogenesis of AMD [34]. The photoreceptor cells and retinal pigment epithelium (RPE) cells in the retina need to be metabolized in a high oxygen environment to fully exert their physiological functions, but at the same time, this will lead to a large accumulation of reactive oxygen species (ROS). In addition, due to the presence of a large number of unsaturated fatty acids and photoreceptor cytochromes, the macular region exhibits characteristics including high oxygen consumption and sensitivity to light radiation, which will further generate ROS [34, 35]. The increased level of ROS not only directly damages the components of RPE cells and photoreceptor cells such as proteins and lipids, thereby impairing their physiological functions, but also stimulates RPE cells to produce VEGF and hypoxia-inducible factor 1 to stimulate CNV generation [34, 36].

Several studies have found that acupuncture has the effect of maintaining redox homeostasis, which is achieved by modulating the imbalance between pro-oxidants and antioxidants [37]. Acupuncture can reduce oxidative stress and injury by inhibiting the production of ROS, reducing the ratio of the redox state of plasma glutathione/oxidized glutathione, and increasing the expression of redox effector [32, 33, 38, 39]. In addition, the regulatory effects of acupuncture on autophagy [40], inflammatory factors [41], and complement levels [42] may also play a role in delaying the progression of AMD.

Some studies have reported positive effects of acupuncture in AMD patients. Krenn et al. found in an observational study that acupuncture may have a positive effect on the vision of AMD patients [11]. Li et al. found that acupuncture improved the visual acuity of patients and reduced the levels of the macular nerve fiber layer, retinal neuroepithelium layer, pigment epithelium, and choroid capillary composite layer. Further, a 3-month follow-up showed that the improvement effect on visual acuity and macular retinal structure was still be maintained [12]. Other studies have also reported positive effects of acupuncture in patients with dry or wet AMD, including improving visual prognosis [43–45], alleviating ocular symptoms (e.g., visual distortion, blurred vision, visual fatigue, shadow occlusion) [43, 44], and enhancing the quality of life of the patients [45]. Similarly, our meta-analysis showed that acupuncture improved clinical efficacy rates and BCVA in AMD patients, and sensitivity analysis showed stability for both outcomes. However, when limiting the duration of intervention to 20–50 days, we did not observe significant improvements in BCVA and CMT in the acupuncture group compared to the control group, suggesting that prolonged intervention may be necessary for the efficacy of acupuncture. In addition, a subgroup analysis of wet AMD patients showed no improvement in BCVA and CMT with acupuncture, despite its ability to improve clinical efficiency in patients with wet AMD. It should be noted that the limitation of the number of studies in the subgroup analysis reduces the certainty of this conclusion. None of the studies mentioned the follow-up of patients' visual acuity; therefore, we cannot provide evidence to support the effects of acupuncture on the long-term vision prognosis of patients.

Results from a meta-analysis of 3 studies showed that compared with the control, acupuncture reduced the CMT more in the AMD patients. However, the lack of included literature and the high heterogeneity of outcomes greatly limit the certainty of this evidence. CMT is an important indicator of OCT for observing the morphological structure and pathological changes of the macular area. An increase in CMT is related to the deterioration of retinal function and can damage visual acuity in AMD patients [46–49]. Acupuncture can reduce CMT, which may be related to the expansion of the surrounding blood vessels, thereby improving the microcirculation in the macular region [16–18]. This microcirculation can not only improve the nutrient supply to the working cells in the fundus [50] but also promote the absorption of hemorrhage and edema caused by CNV to restore the normal shape and function of the macular area. Only two studies reported side effects [17, 28]. The failure to mention the qualifications of the acupuncturists and the small number of research samples make their finding extremely uncertain. We noted that one of the studies reported 10 cases of bleeding at the site of acupuncture [28]. After further communication with the author, the bleeding occurred when the needle was withdrawn and could be stopped after a cotton swab was pressed against the bleeding site for around 10 seconds. It is necessary to standardize the extraction process of acupuncture needles. In reports of acupuncture treatment of other eye diseases, acupuncture was a relatively safe treatment method [51–53], and its safety in application to AMD treatment still needs further research.

All the included studies were conducted in China, so the global inference is limited. The population involved in this study was over 50 years old; considering that cases of AMD between the ages of 40 and 50 are not uncommon [54], we cannot assume that our results

apply to all age groups. In addition, we reviewed the related technologies for acupuncture treatment in AMD. None of the studies provided detailed information on the acupuncture practitioners. Our senior acupuncture review authors (WQP and ZJ) found that most of the studies (88.9%) selected appropriate acupoints because these acupoints were used by professionally trained and experienced clinicians for a long time. Seven studies used a sufficient frequency and treatment time. Nine studies mentioned specific acupuncture techniques, which can ensure the reproducibility of the acupuncture implementation. None of the included studies guaranteed that participants or acupuncturists had been blinded successfully. The quality of evidence was limited by the high risk of co-intervention bias (performance bias), failure to use intentional analysis (attrition bias), and the inconsistency (high heterogeneity; $I^2 = 89\%$, for BCVA; $I^2 = 67\%$, for CMT) and imprecision (335 samples for BCVA, 201 samples for CMT) of studies involving BCVA and CMT. Therefore, no evidence was highly certain. The quality of our evidence ranged from "low" to "very low" according to the GRADE evaluation system (S2 Table in S1 Appendix).

### 4.2 Limitations

First, the quality of included studies is of concern, with many studies failing to mention allocation concealment, or blinding of participants or personnel, limiting our understanding of the evidence. Second, although we implemented an adequate and detailed search strategy, the possibility of publication bias cannot be ruled out, which means that some result values may be amplified, especially in the presence of selective reporting bias in some included studies. Also, most studies used clinical efficacy rates as an outcome measure rather than BCVA or other international standard vision outcomes, failing to thoroughly assess the efficacy of acupuncture.

### 5. Conclusion

Limited evidence suggests that patients with AMD may benefit from acupuncture, especially those with dry AMD. Considering the potential of acupuncture treatment, it is necessary to carry out a rigorously designed RCT to verify its efficacy.

### Supporting information

**S1 Checklist. PRISMA 2009 checklist.**
(DOC)

**S1 Appendix.**
(PDF)

### Acknowledgments

All authors of this manuscript would like to express our sincere gratitude to the ophthalmologist of Beijing University of Chinese Medicine Oriental Hospital for reviewing acupuncture techniques.

### Author Contributions

**Conceptualization:** Yuwei Zhao.

**Data curation:** Xueyao Wang.

**Methodology:** Wu Sun, Yuwei Zhao, Liang Liao, Guojun Chao, Jian Zhou.

**Software:** Liang Liao, Xueyao Wang.

**Supervision:** Guojun Chao, Jian Zhou.

**Writing – original draft:** Wu Sun, Yuwei Zhao.

**Writing – review & editing:** Qiping Wei, Guojun Chao, Jian Zhou.

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
