## [Decision Letter · Decision Letter 0]

30 Aug 2022

PONE-D-22-08228Effects of acupuncture on age-related macular degeneration: a systematic review and meta-analysis of randomized controlled trialsPLOS ONE

Dear Dr. Zhou,

Thank you for submitting your manuscript to PLOS ONE. After careful consideration, we feel that it has merit but does not fully meet PLOS ONE’s publication criteria as it currently stands. Therefore, we invite you to submit a revised version of the manuscript that addresses the points raised during the review process.

We look forward to receiving your revised manuscript.

Kind regards,

Ditta Zobor, MD

Academic Editor

PLOS ONE

https://journals.plos.org/plosone/s/file?id=ba62/PLOSOne_formatting_sample_title_authors_affiliations.pdf".

“This study was supported by The National Natural Science Foundation of China (NSFC), No.81874491.”

Please state what role the funders took in the study.  If the funders had no role, please state: ""The funders had no role in study design, data collection and analysis, decision to publish, or preparation of the manuscript.

“This study was supported by The National Natural Science Foundation of China (NSFC), No.81874491.”

“This study was supported by The National Natural Science Foundation of China (NSFC), No.81874491.”

“None”

5. Please amend your authorship list in your manuscript file to include author names.

Reviewers' comments:

Reviewer's Responses to Questions

**Comments to the Author**

1. Is the manuscript technically sound, and do the data support the conclusions?

Reviewer #1: Yes

Reviewer #2: Partly

2. Has the statistical analysis been performed appropriately and rigorously? 

Reviewer #1: Yes

Reviewer #2: Yes

3. Have the authors made all data underlying the findings in their manuscript fully available?

Reviewer #1: Yes

Reviewer #2: No

4. Is the manuscript presented in an intelligible fashion and written in standard English?

Reviewer #1: Yes

Reviewer #2: Yes

5. Review Comments to the Author

Reviewer #1: This meta-analysis aims to review the clinical efficacy of acupuncture in the treatment of age-related macular degeneration (AMD). In fact，there is no high-quality evidence that acupuncture is effective in treating patients with AMD. Analysis showed patients with dry AMD may benefit from acupuncture. Overall, this study has clinical significance. It can provide some evidence for clinical practice and further clinical trial.

Figures should be more clear.

Reviewer #2: Overall, the methodology is acceptable overall. However, there are some limitations as follows.

1. In methods, the criteria for considering studies for the review should move in front of the search strategy. And in 2.3.1, just RCTs is enough, the second half of the sentences could be omit.

2. In 2.3.2, at least one diagnostic criteria should be the references.

3. 2.4, I believe the exclusion criteria is not enough.

4. In the search strategy, supplement 1.1, the search strategies are not rigorous, Published, highly sensitive, validated search filters to identify randomized trials should be considered, not just RCT(publication Type).

5. in 2.5, the duplicate documents should be filtered by software, not human.

6. Some of the methodologies are outdated: risk of bias tools 2.0 should be used; Revman Web should be used, or at least Revman 5.4.

7. In 3.3, as there is no sham acupuncture, how to blind the practioners?

8. In table 3, is the P value for the heterogeneity, or for the effect size RR.

9. In 3.4.4, one study reported 10 of 22 patients in the acupuncture group had bleeding. I assume there are more discussion about the safety of acupunture of AMD, the acupoints, the manipulation, etc..

6. PLOS authors have the option to publish the peer review history of their article (what does this mean?). If published, this will include your full peer review and any attached files.

Reviewer #1: No

Reviewer #2: **Yes: **Chen, Chao

---

## [Author Response · Author response to Decision Letter 0]

6 Sep 2022

Jian Zhou 

Dear Editors and Reviewers: 

Thank you for your letter and the reviewers’ comments concerning our manuscript 

entitled “Effects of acupuncture on age-related macular degeneration: a systematic review and meta-analysis of randomized controlled trials” (Manuscript ID: PONE-D-22-08228). Those comments are all valuable and very helpful for revising and improving our paper, as well as the important guiding significance to our research. We have studied the comments carefully and have made corrections which we hope meet with approval. Revised portions are marked in red on the paper. The main corrections in the paper and the responses to the reviewer’s comments are as flowing: 

Responds to the reviewer’s comments: 

1. Response to comment: adjust for manuscript style.

Response: Thank you for your suggestion. We changed the style based on the style templates.

2. Response to comment: stating the financial disclosure.

Response: Thank you for your suggestion. We added the financial disclosure in the cover letter.

3. Response to comment: remove any funding-related text from the manuscript: 

Response: Thank you for your suggestion. We removed the funding-related text from the manuscript and add it in the cover letter.

4. Response to comment: stating Competing Interests section.

Response: We added the Competing Interests section in the cover letter, thank you.

5. Response to comment: amend authorship list.

Response: We did a relevant adjustment to the authorship list. Thank you for your suggestion. 

Special thanks to you for your good comments！ 

Reviewer #1: 

1. Response to comment: Figures should be more clear.

Response: Thank you very much for your recognition of this paper. We re-adjusted the graphics quality, which is shown in Figure 2. 

Special thanks to you for your good comments！ 

Reviewer #2: 

1. Response to comment: In methods, the criteria for considering studies for the review should move in front of the search strategy. And in 2.3.1, just RCTs is enough, the second half of the sentences could be omitted.

Response: Thank you for your suggestion. We moved the criteria in front of the search strategy and made adjustments to the study type in the inclusion criteria. 

2. Response to comment: add diagnostic criteria.

Response: Thank you for your suggestion. We supplemented the diagnostic criteria for AMD and please refer to them in 2.2.2. Thank you.

3. Response to comment: amend exclusion criteria.

Response: Thank you for your suggestion. We amended the exclusion criteria and please refer to them in 2.3. Thanks for your opinion.

4. Response to comment: modify the search strategies.

Response: Thank you very much for your suggestion. We added "random" or "randomized trials" to the search strategy and did the search again. The number of studies we searched changed, but no new studies were found that met the inclusion criteria. Please refer to the supplementary material and Figure 1. Thanks for your opinion.

5. Response to comment: the duplicate documents should be filtered by software, not humans.

Response: Thank you for your comments. We apologize for not being clear on this part. We used Endnote software for the preliminary deletion of duplicate articles. We have updated this section in 2.5. Thank you. 

6. Response to comment: Some of the methodologies are outdated.

Response: Thank you for kindly remind. We used ROBIS 2 to re-evaluate the quality of the literature, please refer to Figure 2. In addition, we used Revman 5.4 and the results did not change in any way. We have updated the content in the corresponding part. Thank you.

7. Response to comment: In 3.3, as there is no sham acupuncture, how to blind the practitioners?

Response: What we mean by blinding here is that the intervening acupuncturist was unaware that the patient was the researcher. We realized that the description here was inappropriate and removed this content. Thank you for remind.

8. Response to comment: In table 3, is the P value for the heterogeneity, or for the effect size RR.

Response: The P value is for effect size including RR MD or SMD.

9. Response to comment: add more discussion about the safety of acupuncture.

Response: Thank you for your suggestion. We noted that one of the studies reported 10 cases of bleeding at the site of acupuncture. After further communication with the authors, the bleeding occurred when the needle was withdrawn and could be stopped after a cotton swab was pressed against the bleeding site for 10 seconds. We believe that this is mainly caused by the improper operation of acupuncture extraction. In our study, there was insufficient evidence to investigate the safety of acupuncture for AMD due to few studies reporting side effects. Combined with the available evidence, we found no significant side effects of acupuncture. In other meta-analyses of acupuncture in the treatment of ophthalmic diseases [1-3], acupuncture has shown a good safety profile. We have added relevant descriptions to the safety issues of acupuncture, thank you.

Special thanks to you for your good comments！

Other changes: 

Since we have re-performed the research process, we have updated the content of supplementary material 1.1.

We tried our best to improve the manuscript and made some changes in the manuscript. 

These changes will not influence the framework of the paper. And here we did not list the changes but marked them in red in the revised paper. 

We appreciate for Editors/Reviewers’ warm work earnestly and hope that the correction will meet with approval. 

Once again, thank you very much for your comments and suggestions.

---

## [Decision Letter · Decision Letter 1]

8 Mar 2023

Effects of acupuncture on age-related macular degeneration: a systematic review and meta-analysis of randomized controlled trials

PONE-D-22-08228R1

Dear Dr. Zhou,

We’re pleased to inform you that your manuscript has been judged scientifically suitable for publication and will be formally accepted for publication once it meets all outstanding technical requirements.

Kind regards,

Shiying Li, MBBS

Academic Editor

PLOS ONE

Additional Editor Comments (optional):

All comments are responsed, I suggested to accept it.

Reviewers' comments:

Reviewer's Responses to Questions

**Comments to the Author**

1. If the authors have adequately addressed your comments raised in a previous round of review and you feel that this manuscript is now acceptable for publication, you may indicate that here to bypass the “Comments to the Author” section, enter your conflict of interest statement in the “Confidential to Editor” section, and submit your "Accept" recommendation.

Reviewer #3: All comments have been addressed

Reviewer #4: (No Response)

2. Is the manuscript technically sound, and do the data support the conclusions?

Reviewer #3: Partly

Reviewer #4: Yes

3. Has the statistical analysis been performed appropriately and rigorously? 

Reviewer #3: Yes

Reviewer #4: Yes

4. Have the authors made all data underlying the findings in their manuscript fully available?

Reviewer #3: Yes

Reviewer #4: Yes

5. Is the manuscript presented in an intelligible fashion and written in standard English?

Reviewer #3: Yes

Reviewer #4: Yes

6. Review Comments to the Author

Reviewer #3: 1. Line 18~19, please describe the ‘relationship’ more exactly, it would be better to describe what relationship is.

2. Line 36, please add the full name of the ‘RCT’ because this is the first time to be here.

3. Table 1, there are some overlapped number markers (‘Outcomes’), it would be better to make them more clear by adjusting them, for example use 1~4 to replace ‘1234’

4. Line 250~259, when you discussed the ‘TCM theory’, it would be better to add more references to address this question, especially more discussion about why the acupuncture is not effective in treating the patients with AMD.

5. In Discussion, it would be better to discuss the effects of the ‘acupuncture points’, for example BL1.

Reviewer #4: 1.The literature collection process is detailed and clear, but most of the content of the discussion is mainly summary, the review content can be further enriched.

2.Possible bias could be more specific

7. PLOS authors have the option to publish the peer review history of their article (what does this mean?). If published, this will include your full peer review and any attached files.

Reviewer #3: No

Reviewer #4: No

---

## [Editor Report · Acceptance letter]

14 Mar 2023

PONE-D-22-08228R1 

Effects of acupuncture on age-related macular degeneration: a systematic review and meta-analysis of randomized controlled trials 

Dear Dr. Zhou:

I'm pleased to inform you that your manuscript has been deemed suitable for publication in PLOS ONE. Congratulations! Your manuscript is now with our production department. 

Kind regards, 

on behalf of

Dr. Shiying Li 

Academic Editor

PLOS ONE